# Adolescents and young adults are the most undiagnosed of HIV and virally unsuppressed in Eastern and Southern Africa: Pooled analyses from five population-based surveys

Helena Huerga[1]☯*, Jihane Ben Farhat[1]☯, David Maman[1], Nolwenn Conan[1], Gilles Van Cutsem[2,3], Willis Omwoyo[4], Daniela Garone[5], Reinaldo Ortuno Gutierrez[6], Tsitsi Apollo[7], Gordon Okomo[8], Jean-François Etard[1]

1 Department of Field Epidemiology, Epicentre, Paris, France, 2 Southern Africa Medical Unit, Médecins Sans Frontières, Cape Town, South Africa, 3 Centre for Infectious Disease Epidemiology and Research, University of Cape Town, Cape Town, South Africa, 4 AIDS & STI Department, Ministry of Health, Homa Bay, Kenya, 5 Department of Medicine, Médecins Sans Frontières, Harare, Zimbabwe, 6 Department of Medicine, Médecins Sans Frontières, Nsanje, Malawi, 7 Department of HIV/STIs, Ministry of Health and Child Care, Harare, Zimbabwe, 8 Department of Medicine, Ministry of Health, Homa Bay, Kenya

☯ These authors contributed equally to this work.
* helena.huerga@epicentre.msf.org

**Data Availability Statement:** Data collected for the study contain sensitive patient information. Data

## Abstract

Age and gender disparities within the HIV cascade of care are critical to focus interventions efficiently. We assessed gender-age groups at the highest probability of unfavorable outcomes in the HIV cascade in five HIV prevalent settings. We performed pooled data analyses from population-based surveys conducted in Kenya, South Africa, Malawi and Zimbabwe between 2012 and 2016. Individuals aged 15–59 years were eligible. Participants were tested for HIV and viral load was measured. The HIV cascade outcomes and the probability of being undiagnosed, untreated among those diagnosed, and virally unsuppressed ($\geq$1,000 copies/mL) among those treated were assessed for several age-gender groups. Among 26,743 participants, 5,221 (19.5%) were HIV-positive (69.9% women, median age 36 years). Of them, 72.8% were previously diagnosed and 56.7% virally suppressed (88.5% among those treated). Among individuals 15–24 years, 51.5% were diagnosed *vs* 83.0% among 45–59 years, p<0.001. Among 15–24 years diagnosed, 60.6% were treated *vs* 86.5% among 45–59 years, p<0.001. Among 15–24 years treated, 77.9% were virally suppressed *vs* 92.0% among 45–59 years, p<0.001. Among all HIV-positive, viral suppression was 32.9% in 15–24 years, 47.9% in 25–34 years, 64.9% in 35–44 years, 70.6% in 45–59 years. Men were less diagnosed than women (65.2% *vs* 76.0%, p <0.001). Treatment among diagnosed and viral suppression among treated was not different by gender. Compared to women 45–59 years, young people had a higher probability of being undiagnosed (men 15–24 years OR: 37.9, women 15–24 years OR: 12.2), untreated (men 15–24 years OR:2.2, women 15–24 years OR: 5.7) and virally unsuppressed (men 15–24 years OR: 1.6, women 15–24 years OR: 6.6). In these five Eastern and Southern Africa settings, adolescents and young adults had the largest gaps in the HIV cascade. They were less

will be made available upon request after manuscript publication. Data will include individual deidentified participant data and data dictionary. Requests can be addressed to dpco. archive@epicentre.msf.org. Requests will be examined by a committee of relevant persons involved in the study. The scientific aspects of the proposal as well as the ethical and legal implications of the data sharing will be considered. Data will be shared after approval of the proposal and after signing a data sharing agreement by all parties involved.

**Funding:** This study was financially supported by Médecins Sans Frontières in the form of a project award. There are no relevant grant or award numbers associated with this funding. This study was also financially supported by Médecins Sans Frontières in the form of salaries for authors GVC, DG, and ROG. The specific roles of these authors are articulated in the 'author contributions' section. No additional external funding was received for this study. The funder had no role in study design, data collection and analysis, decision to publish, or preparation of the manuscript.

**Competing interests:** The authors have read the journal's policy and have the following competing interests: GVC, DG, and ROG are employees of Médecins Sans Frontières. This does not alter our adherence to PLOS policies on sharing data and materials. There are no patents, products in development or marketed products associated with this research to declare.

diagnosed, treated, and virally suppressed, than older counterparts. Targeted preventive, testing and treating interventions should be scaled-up.

## Introduction

The Joint United Nations Programme on HIV/AIDS (UNAIDS) has developed the ambitious, initially 90–90–90, and currently 95-95-95 strategy, with the objective to end the AIDS epidemic as a public health threat by 2030 by achieving, among others measures, the following three HIV care targets: 95% of all people living with HIV know their status; 95% of all people diagnosed with HIV receive sustained antiretroviral therapy (ART); and 95% of all people on ART are virally suppressed [1]. The process from initial diagnosis to ART initiation and finally viral suppression is often referred to as the HIV diagnosis and treatment cascade.

There are reports showing poorer outcomes on the HIV testing and treatment cascade (including the 90-90-90 indicators) at younger ages [2–6]. However, a large study conducted in four low-income countries did not find an effect of young age on HIV testing [7]. There is also substantial evidence of a gender differential within the HIV cascade, with men having lower rates of HIV diagnosis and treatment and higher mortality [2,8–10]. Designing specific interventions to address gender inequities has been recommended but translation into operational programs on a large scale is still pending [11,12].

The main effects of these two major determinants, and their potential interactions, on the HIV diagnosis and treatment cascade outcomes need to be translated into prevention, testing and treatment strategies to the benefit of the groups most in need. To tailor public policy or guidelines toward these groups, it is imperative to evaluate these effects at the operational level in high HIV prevalence countries. Population-based surveys provide accurate and valid information on HIV diagnosis, HIV testing and treatment coverage in the community to fulfill these objectives. From 2012 to 2016, Médecins Sans Frontières (MSF), in collaboration with Ministries of Health (MoH), conducted five large HIV population-based surveys in four Eastern and Southern African countries with high HIV prevalence and long-term HIV-AIDS programs. We aimed to assess the age-gender groups with the lowest HIV diagnosis and treatment coverage and at the highest probability of unfavorable outcomes in the HIV cascade in five highly prevalent HIV settings.

## Methods

### Study design and population

We performed pooled analyses that included data from five population-based cross-sectional surveys conducted in Ndhiwa (Kenya) in 2012, KwaZulu-Natal (South Africa) and Chiradzulu (Malawi) in 2013, and Nsanje (Malawi) and Gutu (Zimbabwe) in 2016. In each site, a multi-stage cluster probability sampling was used for the selection of households. For each survey, all individuals aged 15–59 years (Ndhiwa, KwaZulu-Natal, Chiradzulu) and aged 15 or above (Nsanje, Gutu) living in the selected households at survey time were invited to participate to the study. The surveys were conducted over three months in each of the sites. More details about the methods used in each of the surveys can be found in the dedicated papers [13–17]. The current analysis includes all surveyed individuals aged 15–59 years.

### Study settings

The five surveys were conducted in high HIV prevalent areas of Eastern and Southern Africa. The areas were well delimited administratively (sub-counties, districts, or municipalities) and

population ranged from 120,000 to 300,000 inhabitants. In all areas, MSF and MoH supported HIV/AIDS programs including HIV testing and treatment. Additional information about the study settings and the HIV/AIDS programs can be found in the S1 Text and the S1 Table.

### Surveys' procedures

Demographic, socio-economic, behavioral, and clinical data were collected through face-to-face interviews using dedicated questionnaires. All participants were tested for HIV at home, by certified lay counsellors using a serial rapid testing algorithm according to respective national guidelines. Pre- and post-test counselling time were offered to all participants. Awareness of HIV status and ART use were self-reported. In KwaZulu-Natal qualitative testing for presence of antiretrovirals in blood was also performed. Individuals who tested positive for HIV had a venous blood sample collected for viral load [18]. Viral load measurement was performed for all individuals who tested positive for HIV, regardless of their ART status.

### Statistical analyses

The demographic characteristics of HIV-positive individuals surveyed as well as the HIV testing and treatment cascade were described for each site and globally. To evaluate the HIV diagnosis and treatment coverage and identify the age-gender groups with the highest proportions of unfavorable outcomes in the HIV cascade, three main indicators were estimated for the HIV-positive individuals: 1) Diagnosed, defined as the proportion of individuals who declared being diagnosed with HIV prior to the survey among all HIV-positive individuals; 2) Treated, defined as the proportion of individuals receiving ART among individuals aware of their HIV status; 3) Viral load below 1,000 copies/mL (referred as virally suppressed), defined as the proportion of individuals with viral load below 1,000 copies/mL among individuals on ART. In addition, the overall proportion of HIV-positive individuals with viral load below 1,000 copies/mL was also calculated.

To describe the HIV testing and treatment cascade across age and gender groups, we calculated the proportion of individuals diagnosed, treated and with viral load below 1,000 copies/mL according to their age (15–24, 25–34, 35–44, 45–59 years) and gender (women, men) separately. In addition, we estimated the odds of being undiagnosed, untreated and with viral load ≥1,000 copies/mL for several combined age-gender groups using the group of women 45–59 years as reference, in univariate logistic regression models.

Descriptive analyses were performed using frequency and percentage for categorical variables and using median and interquartile range (IQR) for continuous variables. All statistical analyses were adjusted for survey site, cluster and household level using multi-level mixed effect model. Data were analyzed using Stata version 16 (Stata corp., College Station, Texas, USA).

### Ethical issues

All adult participants provided written informed consent. Participants aged 15–17 years provided assent and their parents, guardians or caregivers provided written informed consent for them. The study protocols were approved by the local ethics review boards in each country and the central institutional ethics review boards.

Additional information on the study methods including the names of the ethics review boards can be found in the Supplementary file.

**Table 1. Eligible and included participants in the surveys and in the pooled analyses, by gender, in the 5 surveys.**

|  | Ndhiwa Kenya | Eshowe South Africa | Chiradzulu Malawi | Nsanje Malawi | Gutu Zimbabwe | Overall |
|---|---|---|---|---|---|---|
| **Eligible for survey[1], N** | **6833** | **6688** | **8277** | **5315** | **5439** | **32552** |
| **Included in survey, n** | 6076 | 5649 | 7269 | 4840 | 4979 | 28813 |
| **Inclusion rate, %** | 88.9 | 84.5 | 87.8 | 91.1 | 91.5 | 88.5 |
| **Included in pooled analyses[2], N** |  |  |  |  |  |  |
| All | 6076 | 5649 | 7269 | 4016 | 3733 | 26743 |
| Women | 3755 | 3518 | 4274 | 2278 | 2181 | 16006 |
| Men | 2321 | 2131 | 2995 | 1738 | 1552 | 10737 |
| **HIV positive, n** |  |  |  |  |  |  |
| All | 1457 | 1423 | 1233 | 525 | 583 | 5221 |
| Women | 1000 | 1085 | 839 | 350 | 375 | 3649 |
| Men | 457 | 338 | 394 | 175 | 208 | 1572 |
| **HIV prevalence, %** |  |  |  |  |  |  |
| All | 24.1 | 25.2 | 17.0 | 12.1 | 13.6 | 19.5 |
| Women | 26.7 | 30.9 | 19.7 | 15.4 | 17.2 | 22.8 |
| Men | 19.8 | 15.9 | 13.0 | 10.1 | 13.4 | 14.6 |

1. Eligibility and inclusion age criteria was 15–59 years for Ndhiwa, Eshowe and Chirdazulu, and 15 years or more for Gutu and Nsanje.

2. Included in pooled analyses if aged 15–59 years.

## Results

### Study population

A total of 14,639 households were surveyed: 3,300 in Ndhiwa, 2,377 in KwaZulu-Natal, 4,115 in Chiradzulu, 2,443 in Nsanje and 2,404 in Gutu. In the households surveyed, 32,552 individuals were eligible and 28,813 (88.5%) were included (Table 1).

In total, 26,743 participants aged 15–59 years were included in the pooled analyses. Of them, 5,221 (19.5%) were HIV-positive. Among HIV-positive participants, 3,649 (69.9%) were women and median age was 36 [IQR: 28–44] years (Table 2). S2 Table in the supplementary file shows the participants' median age by gender. HIV prevalence was higher for women than for men in all surveys and increased with age, plateauing between 30 and 45 years among women and between 35 and 50 years among men (S1 Fig).

**Table 2. Age and gender of the HIV-positive participants in the 5 surveys.**

|  | Ndhiwa Kenya | Eshowe South Africa | Chiradzulu Malawi | Nsanje Malawi | Gutu Zimbabwe | Overall |
|---|---|---|---|---|---|---|
|  | n (%) | n (%) | n (%) | n (%) | n (%) | n (%) |
| **N** | **1457** | **1423** | **1233** | **525** | **583** | **5221** |
| **Age (years)** |  |  |  |  |  |  |
| All, median (IQR) | 34 [27-43] | 34 [27-42] | 36 [31-44] | 37 [30-45] | 41 [33-47] | 36 [28-44] |
| 15-24 | 239 (16.4) | 250 (17.6) | 95 (7.7) | 56 (10.7) | 56 (9.6) | 696 (16.3) |
| 25-34 | 524 (36.0) | 497 (34.9) | 403 (32.7) | 140 (26.7) | 105 (18.0) | 1669 (32.0) |
| 35-44 | 384 (26.4) | 386 (27.1) | 452 (36.7) | 197 (37.5) | 226 (38.8) | 1645 (31.5) |
| 45-59 | 310 (21.3) | 290 (20.4) | 283 (23.0) | 132 (25.1) | 196 (33.6) | 1211 (23.2) |
| **Gender** |  |  |  |  |  |  |
| Women | 1000 (68.3) | 1085 (76.3) | 839 (68.1) | 350 (66.7) | 375 (64.3) | 3649 (69.9) |
| Men | 457 (31.4) | 338 (23.8) | 394 (32.0) | 175 (33.3) | 208 (35.7) | 1572 (30.1) |

**Table 3. HIV testing and treatment cascade of care per age, gender and survey site.**

| | Diagnosed among HIV-positive | On ART among diagnosed | Virally suppressed among treated | Virally suppressed among HIV-positive |
|---|---|---|---|---|
| | n (%) | n (%) | n (%) | n (%) |
| N | 5221 | 3760 | 2951 | 5221 |
| **All** | 3760 (72.8) | 2951 (78.7) | 2568 (88.5) | 2825 (56.7) |
| **Age group (years)** | | | | |
| 15–24 | 351 (51.5) | 212 (60.6) | 162 (77.9) | 215 (32.9) |
| 25–34 | 1121 (67.9) | 791 (70.7) | 665 (84.8) | 756 (47.9) |
| 35–44 | 1291 (79.0) | 1087 (84.5) | 960 (90.4) | 1026 (64.9) |
| 45–59 | 997 (83.0) | 861 (86.5) | 781 (92.0) | 828 (70.6) |
| **Gender** | | | | |
| Women | 2747 (76.0) | 2146 (78.3) | 1869 (88.5) | 2065 (59.0) |
| Men | 1013 (65.2) | 805 (79.6) | 699 (88.5) | 760 (51.3) |
| **Site** | | | | |
| Ndhiwa, Kenya | 858 (59.6) | 585 (68.2) | 477 (82.5) | 545 (39.9) |
| Eshowe, South Africa | 1065 (75.2) | 741 (70.0) | 689 (93.1) | 796 (57.1) |
| Chiradzulu, Malawi | 930 (76.7) | 772 (83.0) | 689 (90.8) | 726 (61.8) |
| Nsanje, Malawi | 404 (77.5) | 378 (94.0) | 329 (89.2) | 358 (72.3) |
| Gutu, Zimbabwe | 503 (86.7) | 475 (94.6) | 384 (84.0) | 400 (72.2) |

## HIV cascade of care by gender and age

Of all HIV-positive individuals, 72.8% (3760/5221) were diagnosed, 53.5% (2951/5221) were on ART and 56.7% (2825/4984) had viral load <1,000 copies/mL. Among the individuals diagnosed, 78.7% were on ART and among those on ART, 88.5% had viral load <1,000 copies/mL. Table 3 shows the results of the HIV testing and treatment cascade per age group, gender, and site.

Young individuals were less diagnosed than older counterparts (51.5% in 15–24 years *vs* 83.0% in 45–59 years, p<0.001). In addition, young people 15–34 years accounted for 61.1% (860/1408) of the total individuals undiagnosed. HIV diagnosis increased with age among women and men, plateauing beyond 35 years old in women (Fig 1). With respect to gender,

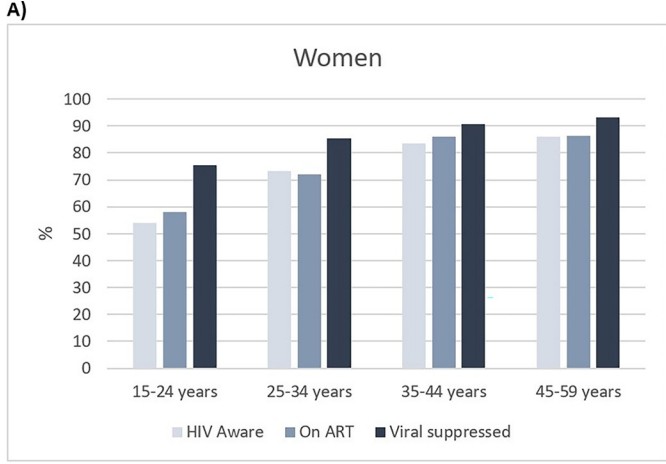
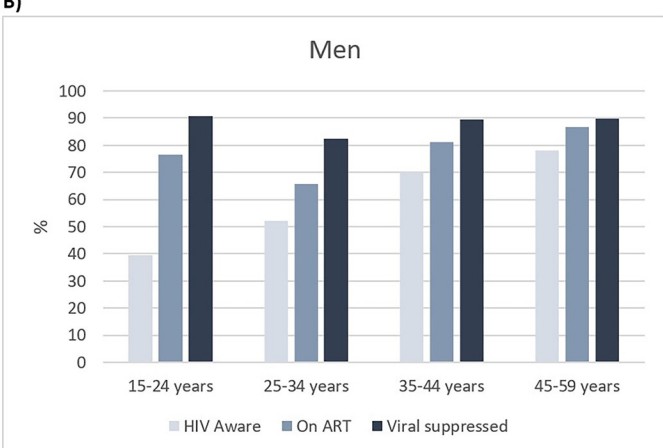

**Fig 1.** HIV testing and treatment cascade per age group among women, N = 3649 (panel A), and men, N = 1572 (panel B); HIV diagnosis among all HIV-positive, treated among those diagnosed, viral load <1000 copies/mL among treated.

men were proportionally less diagnosed than women (65.2% *vs* 76.0% respectively, p<0.001). However, as HIV-positive women were more represented in the surveyed population (due to the higher HIV prevalence in this group), women represented 61.7% (902/1461) of the total individuals undiagnosed.

Treatment coverage among participants diagnosed and viral load <1,000 copies/mL among participants on ART were lower in young individuals compared to those older (60.6% treated among 15–24 years *vs* 86.5% among 45–59 years, p<0.001; 77.9% viral load <1,000 copies/mL among 15–24 years *vs* 92.0% among 45–59 years, p<0.001). Adolescent girls and young women (15–24 years) on ART were less virally suppressed than adolescent boys and young men (15–24 years): 75.4% (132/175) vs 90.9% (30/33), p = 0.05. However, overall, by gender, there was no difference on treatment coverage among those diagnosed (78.3% in women *vs* 79.6% in men) or viral load <1000 copies/mL among those treated (88.5% in women and 88.5% in men).

With respect to viral load <1,000 copies/mL among all HIV-positive individuals, HIV viral suppression was very low in the youngest individuals (32.9% in 15–24 years) and increased with age (47.9% in 25–34 years, 64.9% in 35–44 years, 70.6% in 45–59 years). Young people aged 15–34 years accounted for 58.4% (1260/2159) of all individuals with viral load ≥1,000 copies/mL while this age group represented 48.3% of the total HIV-positive population. There was no difference by gender in the proportion of viral suppression in adolescent girls and young women, and adolescent boys and young men: 32.7% (177/541) vs 33.6% (38/113), respectively, p = 0.85. However, viral suppression among all HIV-positive individuals was slightly lower among HIV-positive men compared to HIV-positive women (51.3% vs 59.0% respectively, p<0.001).

## Likelihood of being undiagnosed, untreated, and virally unsuppressed in combined gender-age groups

Men aged 15–24 years was the group with the highest likelihood (OR = 37.9) of being undiagnosed compared to women aged 45–59 years (Table 4). Women aged 15–24 years and men aged 25–34 years showed 12- and 15-fold increased odds of being undiagnosed, respectively. Women aged 25–34 years and men of older ages 35–59 years also had a higher likelihood of being undiagnosed compared to women 45–59 years though at a lower level. S2 Fig in the supplementary file shows the logarithm of the odds ratios.

Regarding the likelihood of being untreated among those diagnosed, women and men aged 15–34 years were the groups with the highest probability (2 to 6-fold higher odds) compared to women aged 45–59 years. These gender-age groups were also the ones with the highest probability of having viral load ≥1,000 copies/mL among those treated, except men 15–24 years for whom few individuals were included in this analysis.

## Discussion

In this large, pooled analysis of five community-based household surveys, adolescents and young adults had the lowest coverage in the HIV testing and treatment cascade, and the highest probability of being undiagnosed, untreated and virally unsuppressed after adjustment for sociodemographic factors and survey site. More than half of the HIV-positive adolescents and individuals aged 15–34 years had a viral load of 1,000 copies/mL or above. Men were less likely to be HIV diagnosed than women. However, there were no differences by gender on treatment coverage among diagnosed or viral suppression among treated. Our results show that age is a key aspect when defining strategies to close gaps on HIV diagnosis and treatment.

**Table 4. Univariate logistic regression analyses of being undiagnosed (N = 1408), untreated among those diagnosed (N = 800) and with viral load ≥1000 copies/mL among those treated (N = 335) for various combined gender-age (years) groups.**

| | N | % | Odds ratio | [95% CI] |
|---|---|---|---|---|
| **Undiagnosed** | | | | |
| Women 45–59 | 105 | 14.1 | Ref | - |
| Men 45–59 | 100 | 21.9 | 2.4 | 1.5–3.7 |
| Women 35–44 | 178 | 16.5 | 1.4 | 1.0–2.0 |
| Men 35–44 | 165 | 29.8 | 4.5 | 2.9–6.9 |
| Women 25–34 | 326 | 26.6 | 3.4 | 2.3–4.9 |
| Men 25–34 | 203 | 47.9 | 15.1 | 9.3–24.6 |
| Women 15–24 | 259 | 46.0 | 12.2 | 7.8–19.1 |
| Men 15–24 | 72 | 60.5 | 37.9 | 18.2–78.6 |
| **Untreated among diagnosed** | | | | |
| Women 45–59 | 87 | 13.6 | Ref | - |
| Men 45–59 | 47 | 13.2 | 1.0 | 0.6–1.5 |
| Women 35–44 | 127 | 14.1 | 1.1 | 0.8–1.5 |
| Men 35–44 | 73 | 18.8 | 1.6 | 1.1–2.4 |
| Women 25–34 | 253 | 28.1 | 2.8 | 2.0–3.8 |
| Men 25–34 | 75 | 34.1 | 4.0 | 2.6–6.2 |
| Women 15–24 | 127 | 41.9 | 5.7 | 3.8–8.5 |
| Men 15–24 | 11 | 23.4 | 2.2 | 1.0–5.1 |
| **Viral load ≥1000 copies/mL among treated** | | | | |
| Women 45–59 | 37 | 6.8 | Ref | - |
| Men 45–59 | 31 | 10.2 | 1.8 | 1.0–3.3 |
| Women 35–44 | 70 | 9.3 | 1.5 | 0.9–2.5 |
| Men 35–44 | 32 | 10.4 | 1.8 | 1.0–3.4 |
| Women 25–34 | 94 | 14.7 | 2.9 | 1.7–4.7 |
| Men 25–34 | 25 | 17.5 | 3.7 | 1.8–7.7 |
| Women 15–24 | 43 | 24.6 | 6.6 | 3.3–13.3 |
| Men 15–24 | 3 | 9.1 | 1.6 | 0.4–7.2 |

The low diagnosis coverage among adolescents and young adults may have several reasons. Evidence from Nigeria and Burundi suggests that stigma is an important factor preventing HIV testing among adolescents and young adults [4,19], some young adults preferring self-testing [20]. Although few in our sample, men 15–24 years had a particularly high likelihood of being unaware of their HIV positive status compared to other age-sex groups. Studies in Lesotho and South Africa have also reported low HIV awareness in young people, highlighting a lower test intake in men compared to women, due to gender perceptions and dynamics [21,22]. In our study, men were less diagnosed than women at all ages. Low HIV testing and treatment among men in Africa is often due to poor utilization of health facilities combined with an anticipated loss of social position, stigma and sexual desirability when diagnosed HIV-positive, promoting a mindset whereby testing when "still healthy" is considered undesirable [23]. However, African women have repeated opportunities to be HIV tested throughout their reproductive life, which likely partially explains the increasing proportion of HIV diagnosis with age among women [24]. Indeed, in our study, women 15–24 years were at a significantly increased probability to be unaware of their HIV status compared to older women. As HIV prevalence in women is higher than in men, an important proportion of the overall number of individuals undiagnosed were women, mostly young women. Due to progress in Prevention

of Mother to Child Transmission (PMTCT), the number of perinatally infected adolescents is expected to decrease in the coming years and the relative proportion of HIV-positive adolescents infected during adolescence would be higher [25]. HIV preventive interventions are key in high incidence groups such as adolescents and young women and in specific groups such as transactional sexual partners. These interventions should be coupled with gender-age adapted strategies to improve HIV diagnosis in young people. Recent recommendations include the use of long-acting injectable cabotegravir for HIV prevention [26]. Cabotegravir has shown to be effective and well tolerated for HIV prevention [27,28]. It is a good alternative for adolescents and adults who do not feel comfortable with oral preventive therapy [29,30].

HIV-diagnosed young men and women had also an increased probability of being untreated and virally unsuppressed while on ART, implying that age differences persist along the HIV treatment cascade despite appropriate HIV diagnosis. There is little specific provision for adolescents within the health systems of most countries, this group often falling in the gap between paediatric and adult services. Problems related to the transition between paediatric and adult care are common and include adolescents' fear of leaving supportive paediatric care, competing demands with higher education, starting employment and moving out of the family home, and insufficient proactive transition planning by health services, all of which may contribute to periods of worsened HIV care outcomes [31]. Temporary interruption of ART has also been identified as a reason for lack of viral suppression with young women and people on ART for less than 2 years being more likely to interrupt ART [32]. Therefore, adolescents and young people would benefit from interventions focused on engagement and retention on HIV treatment and care as well as proper follow-up while on ART. With respect to the HIV viral suppression among all HIV-positive, as younger individuals were less aware of their HIV positivity and therefore had not engaged on HIV care, they were also the group less virally suppressed.

This study has some limitations. ART coverage and viral suppression were influenced by the ART eligibility criteria applied at the time of the survey which were different in each survey site. It is also important to acknowledge that we did not specifically account for potential ART resistance as a reason for the lack of viral suppression in some individuals. ART resistance is a critical factor that can hinder the achievement of viral suppression among HIV-infected individuals. Finally, due to the cross-sectional design of the surveys, only individuals who have survived were included in the study which may have led to an overestimation of the HIV testing and treatment coverage in older ages. However, we do not think that this possible survival bias precludes the findings and conclusions of the study.

The study has also some strengths such as the random sampling with a high participation rate, a large number of HIV-positive individuals included in the analysis, and the vast geographical area covered, with surveys conducted in five sites and four countries. In contrast with programmatic/administrative data which often reflect the gap in access to viral load measurement, this study benefited from an almost complete viral load testing and therefore provided a more precise assessment of viral suppression among treated HIV-positive participants. This reinforces the validity of the findings.

## Conclusions

In these five Eastern and Southern Africa settings, adolescents and young adults had the largest gaps in the HIV diagnosis and treatment cascade. They were less diagnosed, less treated, and less virally suppressed, than older counterparts. Strategies such as rapid or self-testing, school testing, sexual education, and community-based approaches should be scaled-up to increase HIV diagnosis. In addition, same-day test and treat, regular viral load monitoring, and early

detection of drug-resistance should continue to improve treatment among people diagnosed and maintain viral suppression among those treated. Innovative strategies adapted to young people including tools used by adolescents and young adults such as social media or young peers may also help. Finally, preventive interventions are key in this context to decrease new infections in young people.

## Supporting information

**S1 Text. Supplementary methods.**
(DOCX)

**S1 Table. Description of the study sites and the HIV program activities in the surveyed areas at the time of the survey.**
(DOCX)

**S2 Table. Median age by gender of the participants included in the pooled analyses in the 5 surveyed sites.**
(DOCX)

**S1 Fig. HIV prevalence by age group in women and men in the 5 surveys.**
(TIFF)

**S2 Fig. Logarithm of odds ratios for the risk of being undiagnosed, untreated and with viral load $\geq$ 1000 copies/mL for various age and gender groups (reference group: Women 45–59 years old).**
(TIFF)

## Author Contributions

**Conceptualization:** Helena Huerga, Jean-François Etard.

**Formal analysis:** Jihane Ben Farhat.

**Investigation:** Helena Huerga, David Maman, Nolwenn Conan, Gilles Van Cutsem, Daniela Garone, Reinaldo Ortuno Gutierrez.

**Methodology:** Helena Huerga, Jean-François Etard.

**Supervision:** Helena Huerga, David Maman, Nolwenn Conan, Gilles Van Cutsem, Willis Omwoyo, Daniela Garone, Reinaldo Ortuno Gutierrez, Tsitsi Apollo, Gordon Okomo.

**Validation:** Helena Huerga.

**Writing – original draft:** Helena Huerga, Jihane Ben Farhat.

**Writing – review & editing:** Helena Huerga, Jihane Ben Farhat, David Maman, Nolwenn Conan, Gilles Van Cutsem, Willis Omwoyo, Daniela Garone, Reinaldo Ortuno Gutierrez, Tsitsi Apollo, Gordon Okomo, Jean-François Etard.

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
