## [Decision Letter · Decision Letter 0]

10 Jul 2023

PGPH-D-23-00391

Adolescents and young adults are the most undiagnosed of HIV and virally unsuppressed in Eastern and Southern Africa: pooled analyses from five population-based surveys.

Dear Dr. Huerga,

Thank you for submitting your manuscript to PLOS Global Public Health. After careful consideration, we feel that it has merit but does not fully meet PLOS Global Public Health’s publication criteria as it currently stands. Therefore, we invite you to submit a revised version of the manuscript that addresses the points raised during the review process.

The reviewers found this to be a strong potential contribution to the literature, and suggested a number of clarifications, including revisions to the introduction, to improve it.

We look forward to receiving your revised manuscript.

Kind regards,

Hannah Hogan Leslie, PhD

Academic Editor

Journal Requirements:

Additional Editor Comments (if provided):

Reviewers' comments:

Reviewer's Responses to Questions

**Comments to the Author**

1. Does this manuscript meet PLOS Global Public Health’s publication criteria? Is the manuscript technically sound, and do the data support the conclusions? The manuscript must describe methodologically and ethically rigorous research with conclusions that are appropriately drawn based on the data presented.

Reviewer #1: Yes

Reviewer #2: Yes

Reviewer #3: Yes

2. Has the statistical analysis been performed appropriately and rigorously?

Reviewer #1: No

Reviewer #2: Yes

Reviewer #3: Yes

3. Have the authors made all data underlying the findings in their manuscript fully available (please refer to the Data Availability Statement at the start of the manuscript PDF file)?

Reviewer #1: Yes

Reviewer #2: Yes

Reviewer #3: Yes

4. Is the manuscript presented in an intelligible fashion and written in standard English?

Reviewer #1: Yes

Reviewer #2: Yes

Reviewer #3: Yes

5. Review Comments to the Author

Reviewer #1: This is a well written paper even though for the four sites, study was conducted before the latest WHO eligibility guidelines were implemented.

However, you mentioned that viral load measurement was performed regardless of ART status, it would be great to also see the breakdown of the VLS status for the undiagnosed study participants especially since your study title speaks to the undiagnosed and non-virally suppressed.

.

Reviewer #2: My review has been added as an attachment [Editor note: copied below instead of attached]; however, I would add that the paper is an important addition to the literature on the treatment cascade of adolescents and young adults spanning 5 settings in four countries, and with a very large sample size. The attached comments should therefore not negate the contribution this piece will make to our knowledge on the subject.

**Abstract**:

Line 47 - 72.8% etc were diagnosed – previously or newly diagnosed? Not clear in abstract but becomes clear in the methods; some readers do not go beyond abstractLine 61 – treating or treatment?

**Introduction**:

Lines 71-72 – May want to stick to the focus of the paper - AYALines 72-74 - Make the case for age differentials first - the focus of the paper (and title), then by gender
Lines 75-71 - Provide background of AYA vs older adults in the 3 key outcomes in the HIV treatment cascade, not just testingLine 77 - Gender does not come out in the title so a reader is led to believe the paper is just about ageOverall, the background does not convincingly or clearly make the case for the paper. Need to tighten this to make the reading flow

**Methods**:

Somewhere under Study Design, consider stating the objective(s) of the analysis in this sectionLine 106 - respective national guidelines to indicate it's not a uniform guideline in all countiesLines 114-115 - This is explained here but not clear in abstractLines 116-119 – Number doesn't tell much. consider just proportion

**Results**:

Lines 209-2013 - Shouldn't this summary be about those undiagnosed. ConfusingOverall, consider also comparing viral load between diagnosed and undiagnosed by age group so we see entry point by age – e.g., are  AYA still performing worse? Another analysis that would be on interest is comparing AGYW with ABYM

**Discussion**:

Lines 218-219 - important reinforcement of risk of AYALine 234 -life/lives?Line 238 - sounds hanging. consider adding .. young 'women'Line 263 - do not (not don’t)Line 269 - …including untreated patients… in discussion. Besides reporting combined data occasionally, the results are mostly on diagnosed and very little in undiagnosed = untreated. See other comments aboveLines 275-276 - Repeated - already stated in 1st part of this sentenceLine 276 -  “Therefore, interventions adapted to these groups should be urgently enhanced”  Sentence does not add value – consider dropping

Reviewer #3: This is an important study given the increasing numbers of adolescents living with HIV and continued gender and age disparities along the HIV continuum of care. The authors attempt to address this critical time by conducting a pooled data analyses of population-based surveys in Kenya, South Africa, Malawi, and Zimbabwe to evaluate for gender- and age-specific trends along the HIV continuum of care. The study found that adolescents and young adults ages 15-24 years old were less likely to know their HIV diagnosis, be on treatment, and virally suppressed compared to older age groups.

Overall, the interpretation of the findings are good. The major strengths of this manuscript are the focus on a population (adolescents and young adults living with HIV in Sub-Saharan Africa) that has high rates of morbidity and mortality at each point of the continuum of care, as well as a large sample size with HIV testing and viral load testing, that supports your discussion. However, a few additional points to consider:

1. Introduction: Good overview of gender/age disparities along HIV continuum of care, however I would add in a few sentences about why you chose to study these particular geographic regions (add in some more epidemiological information about HIV epidemic in Sub-Saharan Africa)

2. Methods: Your statistical methods are well described. However, please expand on what you mean by “In KwaZulu-Natal ART testing in blood was also performed” (line 108). Did you check for ART resistance? Did you for specific drug levels of medications?

3. Discussion: Please expand on past studies that highlight cabotegravir for HIV prevention in adolescents / young adults and why this could be a strategy to help decrease HIV in this population (acceptance/feasibility/uptake studies in this age group?) (line 244)

4. Limitations: I would also mention that study does not account for potential ART resistance as a reason for lack of viral suppression

5. Please review for grammar.

- For example, acronym MoH for Ministries of Health in line 81 is later referred to as MOH in line 100 (make sure they are the same capitalization).

- Sentence on line 106 is incomplete (ends with “included,” but doesn’t state what was included?).

- Sentence on lines 234-235 is difficult to read, please try to reword: “However, African women through their reproductive live have repeated opportunities to be HIV tested which probably explains partially the increasing proportion of HIV diagnosis with age.” � could consider changing to “However, African women have repeated opportunities to be HIV tested throughout their reproductive lives, which likely partially explains the increasing proportion of HIV diagnosis with age among women.”

- Make sure all acronyms are defined (PMTCT in line 239 should have a definition the first time it is used)

6. PLOS authors have the option to publish the peer review history of their article (what does this mean?). If published, this will include your full peer review and any attached files.

**Do you want your identity to be public for this peer review?** For information about this choice, including consent withdrawal, please see our Privacy Policy.

Reviewer #1: No

Reviewer #2: No

Reviewer #3: No

---

## [Decision Letter · Decision Letter 1]

11 Oct 2023

PGPH-D-23-00391R1

Adolescents and young adults are the most undiagnosed of HIV and virally unsuppressed in Eastern and Southern Africa: pooled analyses from five population-based surveys.

Dear Dr. Huerga,

Thank you for submitting your manuscript to PLOS Global Public Health. After careful consideration, we feel that it has merit but does not fully meet PLOS Global Public Health’s publication criteria as it currently stands. Therefore, we invite you to submit a revised version of the manuscript that addresses the points raised during the review process. The revised manuscript has addressed the reviewers' concerns and will be an important addition to the literature. Please see below remaining issues with the statistical approach and terminology that must be addressed before the paper can proceed to publication.

We look forward to receiving your revised manuscript.

Kind regards,

Hannah Hogan Leslie, PhD

Academic Editor

Journal Requirements:

Additional Editor Comments (if provided):

Thank you for addressing the reviewers' comments. This manuscript will be a valuable contribution to the literature.

Please note further revisions requested to bring the article into compliance with the journal's statistical policies and recommendations:

- please clarify the analysis is intended to be descriptive, and reword sentences such as 'To study the effect of age and gender in the HIV testing and treatment cascade' to more accurate language such as 'To describe the HIV testing and treatment cascade across age and gender groups'

- please remove all mention of 'risk' in reference to odds ratios (or change the modeling approach to log binomial to calculate risk ratios instead of odds ratios)

- please strongly consider removing the adjusted multivariable modeling, which detracts rather than adds to this highly informative study, as the adjustment factors are much more likely to be mediating factors than confounders of the relationship between age/gender and these outcomes. The value of this study is in identifying the least served groups as laid out in the aims statement at the end of the introduction; the multivariable analysis is not in keeping with this aim. Please see the following paper for further details if helpful: Conroy S, Murray EJ. Let the question determine the methods: descriptive epidemiology done right. Br J Cancer. 2020;123(9):1351-1352. doi:10.1038/s41416-020-1019-z

Reviewers' comments:

Reviewer's Responses to Questions

**Comments to the Author**

1. If the authors have adequately addressed your comments raised in a previous round of review and you feel that this manuscript is now acceptable for publication, you may indicate that here to bypass the “Comments to the Author” section, enter your conflict of interest statement in the “Confidential to Editor” section, and submit your "Accept" recommendation.

Reviewer #3: All comments have been addressed

2. Does this manuscript meet PLOS Global Public Health’s publication criteria? Is the manuscript technically sound, and do the data support the conclusions? The manuscript must describe methodologically and ethically rigorous research with conclusions that are appropriately drawn based on the data presented.

Reviewer #3: Yes

3. Has the statistical analysis been performed appropriately and rigorously?

Reviewer #3: Yes

4. Have the authors made all data underlying the findings in their manuscript fully available (please refer to the Data Availability Statement at the start of the manuscript PDF file)?

Reviewer #3: Yes

5. Is the manuscript presented in an intelligible fashion and written in standard English?

Reviewer #3: Yes

6. Review Comments to the Author

Reviewer #3: Thank you for your thoughtful responses and revisions to the manuscript. All comments have been addressed.

7. PLOS authors have the option to publish the peer review history of their article (what does this mean?). If published, this will include your full peer review and any attached files.

**Do you want your identity to be public for this peer review?** For information about this choice, including consent withdrawal, please see our Privacy Policy.

Reviewer #3: No

---

## [Editor Report · Decision Letter 2]

22 Nov 2023

Adolescents and young adults are the most undiagnosed of HIV and virally unsuppressed in Eastern and Southern Africa: pooled analyses from five population-based surveys.

PGPH-D-23-00391R2

Dear Dr Huerga,

We are pleased to inform you that your manuscript 'Adolescents and young adults are the most undiagnosed of HIV and virally unsuppressed in Eastern and Southern Africa: pooled analyses from five population-based surveys.' has been provisionally accepted for publication in PLOS Global Public Health.

Best regards,

Hannah Hogan Leslie, PhD

Academic Editor